Forecasting biodiversity in breeding birds using best practices

Harris David J. dave@harris-research.me 1
Taylor Shawn D. 2
White Ethan P. 1 3 4
1 Department of Wildlife Ecology and Conservation, University of Florida , Gainesville , FL , United States of America
2 School of Natural Resources and Environment, University of Florida , Gainesville , FL , United States of America
3 Informatics Institute, University of Florida , Gainesville , FL , United States of America
4 Biodiversity Institute, University of Florida , Gainesville , FL , United States of America
Roper James
Electronic publication date: 2018 Feb 8
Publication date: 2018
Volume: 6
Electronic Location ID: e4278
Received 2017 Sep 19; Accepted 2017 Dec 29
Copyright: ©2018 Harris et al.
Copyright year: 2018
Copyright holder: Harris et al.
License: This is an open access article distributed under the terms of the Creative Commons Attribution License, which permits unrestricted use, distribution, reproduction and adaptation in any medium and for any purpose provided that it is properly attributed. For attribution, the original author(s), title, publication source (PeerJ) and either DOI or URL of the article must be cited.
License URL: https://creativecommons.org/licenses/by/4.0/

Keywords: Birds, Breeding bird survey, Species richness, Biodiversity, Forecasting, Prediction, Climate change, Species distribution model, Time series, Space-for-time substitution

Funding: Gordon and Betty Moore Foundation’s Data-Driven Discovery Initiative GBMF4563 This research was supported by the Gordon and Betty Moore Foundation’s Data-Driven Discovery Initiative through Grant GBMF4563 to Ethan P. White. The funders had no role in study design, data collection and analysis, decision to publish, or preparation of the manuscript.

==============================
Biodiversity forecasts are important for conservation, management, and evaluating how well current models characterize natural systems. While the number of forecasts for biodiversity is increasing, there is little information available on how well these forecasts work. Most biodiversity forecasts are not evaluated to determine how well they predict future diversity, fail to account for uncertainty, and do not use time-series data that captures the actual dynamics being studied. We addressed these limitations by using best practices to explore our ability to forecast the species richness of breeding birds in North America. We used hindcasting to evaluate six different modeling approaches for predicting richness. Hindcasts for each method were evaluated annually for a decade at 1,237 sites distributed throughout the continental United States. All models explained more than 50% of the variance in richness, but none of them consistently outperformed a baseline model that predicted constant richness at each site. The best practices implemented in this study directly influenced the forecasts and evaluations. Stacked species distribution models and “naive” forecasts produced poor estimates of uncertainty and accounting for this resulted in these models dropping in the relative performance compared to other models. Accounting for observer effects improved model performance overall, but also changed the rank ordering of models because it did not improve the accuracy of the “naive” model. Considering the forecast horizon revealed that the prediction accuracy decreased across all models as the time horizon of the forecast increased. To facilitate the rapid improvement of biodiversity forecasts, we emphasize the value of specific best practices in making forecasts and evaluating forecasting methods.

Introduction

Forecasting the future state of ecological systems is increasingly important for planning and management, and also for quantitatively evaluating how well ecological models capture the key processes governing natural systems (Clark et al., 2001; Dietze et al., 2018; Houlahan et al., 2017). Forecasts regarding biodiversity are especially important, due to biodiversity’s central role in conservation planning and its sensitivity to anthropogenic effects (Cardinale et al., 2012; Díaz et al., 2015; Tilman et al., 2017). High-profile studies forecasting large biodiversity declines over the coming decades have played a large role in shaping ecologists’ priorities (as well as those of policymakers; e.g., IPCC, 2014), but it is inherently difficult to evaluate such long-term predictions before the projected biodiversity declines have occurred.

Previous efforts to predict future patterns of species richness, and diversity more generally, have focused primarily on building species distributions models (SDMs; Thomas et al., 2004; Thuiller et al., 2011; Urban, 2015). In general, these models describe individual species’ occurrence patterns as functions of the environment. Given forecasts for environmental conditions, these models can predict where each species will occur in the future. These species-level predictions are then combined (“stacked”) to generate forecasts for species richness (e.g., Calabrese et al., 2014). Alternatively, models that directly relate spatial patterns of species richness to environmental conditions have been developed and generally perform equivalently to stacked SDMs (Algar et al., 2009; Distler, Schuetz & Langham, 2015). This approach is sometimes referred to as “macroecological” modeling, because it models the larger-scale pattern (richness) directly (Distler, Schuetz & Langham, 2015).

Despite the emerging interest in forecasting species richness and other aspects of biodiversity (Jetz, Wilcove & Dobson, 2007; Thuiller et al., 2011), little is known about how effectively we can anticipate these dynamics. This is due in part to the long time scales over which many ecological forecasts are applied (and the resulting difficulty in assessing whether the predicted changes occurred; Dietze et al., 2018). What we do know comes from a small number of hindcasting studies, where models are built using data on species occurrence and richness from the past and evaluated on their ability to predict contemporary patterns (e.g., Algar et al., 2009; Distler, Schuetz & Langham, 2015) or historic (Blois et al., 2013; Maguire et al., 2016) periods not used for model fitting. These studies are a valuable first step, but lack several components that are important for developing forecasting models with high predictive accuracy, and for understanding how well different methods can predict the future. These “best practices” for effective forecasting and evaluation (Box 1) broadly involve: (1) expanding the use of data to include biological and environmental time-series (Tredennick et al., 2016); (2) accounting for uncertainty in observations and processes, (Yu, Wong & Hutchinson, 2010; Harris, 2015); and (3) conducting meaningful evaluations of the forecasts by hindcasting, archiving short-term forecasts, and comparing forecasts to baselines to determine whether the forecasts are more accurate than assuming the system is basically static (Dietze et al., 2018).

In this paper, we attempt to forecast the species richness of breeding birds at over 1,200 of sites located throughout North America, while following best practices for ecological forecasting (Box 1). To do this, we combine 32 years of time-series data on bird distributions from annual surveys with monthly time-series of climate data and satellite-based remote-sensing. Contemporary datasets that span a time scale of 30 years or more have only recently become available for large-scale time-series based forecasting. A dataset of this size allows us to model and assess changes a decade or more into the future in the presence of shifts in environmental conditions on par with predicted climate change. We compare traditional distribution modeling based approaches to spatial models of species richness, time-series methods, and two simple baselines that predict constant richness for each site, on average (Fig. 1). All of our forecasting models account for uncertainty and observation error, are evaluated across different time lags using hindcasting, and are publicly archived to allow future assessment. We discuss the implications of using best practices for our understanding of, and confidence in, the resulting forecasts, and how we can continue to build on these approaches to improve ecological forecasting in the future.

Figure 1 Example predictions from six forecasting models for a single site. Data from 1982 through 2003, connected by solid lines, were used for training the models; the remaining points were used for evaluating the models’ forecasts.

In each panel, point estimates for each year are shown with lines; the darker ribbon indicates the 68% prediction interval (one standard deviation of uncertainty), and the lighter ribbon indicates the 95% prediction interval. (A) Single-site models were trained independently on each site’s observed richness values. The first two models (“average” and “naive”) served as baselines. (B) The environmental models were trained to predict richness based on elevation, climate, and NDVI; the environmental models’ predictions change from year to year as environmental conditions change.

Methods

We evaluated six types of forecasting models (Table 1) by dividing the 32 years of data into 22 years of training data and 10 years of data for evaluating forecasts using hindcasting. Here we use definitions from meteorology, where a hindcast is generally any prediction for an event that has already happened, while forecasts are predictions for actual future events (Jolliffe & Stephenson, 2003). We also made long term forecasts by using the full data set for training and making forecasts through the year 2050. For both time frames, we made forecasts using each model with and without correcting for observer effects, as described below.

Table 1 Six forecasting models.

Single-site models were trained site-by-site, without environmental data. Environmental models were trained using all sites together, without information regarding which transects occurred at which site or during which year. Most of the models were trained to predict richness directly. This mirrors the standard application of these techniques. Separate random forest SDMs were fit for each species and used to predict the probability of that species occurring at each site. The species-level probabilities at a site were summed to predict richness. The mistnet JSDM was trained to predict the full species composition at each site, and the number of species in its predictions was used as an estimate of richness.

		Predictors	
Model	Response variable	Site id	Time	Environment	
Single-site models	
Average baseline	richness	✓			
Naive baseline	richness	✓	✓		
Auto-ARIMA	richness	✓	✓		
Environmental models	
GBM richness	richness			✓	
Stacked SDMs	species-level presence			✓	
Mistnet JSDM	species composition			✓	

Data

Richness data

Bird species richness was obtained from the North American Breeding Bird Survey (BBS) (Pardieck et al., 2017) using the Data Retriever Python package (Morris & White, 2013; Senyondo et al., 2017) and rdataretriever R package (McGlinn et al., 2017). BBS observations are three-minute point counts made at 50 fixed locations along a 40 km route. Here we denote each route as a site and summarize richness as the total number of species observed across all 50 locations in each surveyed year. The data was filtered to exclude all nocturnal, cepuscular, and aquatic species (since these species are not well sampled by BBS methods; Hurlbert & White, 2005), as well as unidentified species, and hybrids. All data from surveys that did not meet BBS quality criteria were also excluded.

We used observed richness values from 1982 (the first year of complete environmental data) to 2003 to train the models, and from 2004 to 2013 to test their performance. We only used BBS routes from the contiguous United States (i.e., routes where climate data was available PRISM Climate Group (2004)), and we restricted the analysis to routes that were sampled during 70% of the years in the training period (i.e., routes with at least 16 annual observations). The resulting dataset included 34,494 annual surveys of 1,279 unique sites, and included 385 species. Site-level richness varied from 8 to 91 with an average richness of 51 species.

Past environmental data

Environmental data included a combination of elevation, bioclimatic variables and a remotely sensed vegetation index (the normalized difference vegetation index; NDVI), all of which are known to influence richness and distribution in the BBS data (Kent, Bar-Massada & Carmel, 2014; Hurlbert & Haskell, 2002). For each year in the dataset, we used the 4 km resolution PRISM data (PRISM Climate Group, 2004) to calculate eight bioclimatic variables identified as relevant to bird distributions (Harris, 2015): mean diurnal range, isothermality, max temperature of the warmest month, mean temperature of the wettest quarter, mean temperature of the driest quarter, precipitation seasonality, precipitation of the wettest quarter, and precipitation of the warmest quarter. These variables were calculated for the 12 months leading up to the annual survey (July–June) as opposed to the calendar year. Satellite-derived NDVI, a primary correlate of richness in BBS data (Hurlbert & Haskell, 2002), was obtained from the NDIV3g dataset with an 8 km resolution (Pinzon & Tucker, 2014) and was available from 1981–2013. Average summer (April, May, June) and winter (December, January, Feburary) NDVI values were used as predictors. Elevation was from the SRTM 90 m elevation dataset (Jarvis et al., 2008) obtained using the R package raster (Hijmans, 2016). Because BBS routes are 40-km transects rather than point counts, we used the average value of each environmental variable within a 40 km radius of each BBS route’s starting point.

Future environmental projections

In addition to the analyses presented here, we have also generated and archived long term forecasts from 2014–2050. This will allow future researchers to assess the performance of our six models on longer time horizons as more years of BBS data become available. Precipitation and temperature were forecast using the CMIP5 multi-model ensemble dataset (Brekke et al., 2013). Thirty-seven downscaled model runs (Brekke et al., 2013, see Table S1) using the RCP6.0 scenario were averaged together to create a single ensemble used to calculate the bioclimatic variables for North America. For NDVI, we used the per-site average values from 2000–2013 as a simple forecast. For observer effects (see below), each site was set to have zero observer bias. The predictions have been archived on Zenodo (Harris, White & Taylor, 2017b).

Accounting for observer effects

Observer effects are inherent in large data sets collected by different observers, and are known to occur in BBS (Sauer, Peterjohn & Link, 1994). For each forecasting approach, we trained two versions of the corresponding model: one with corrections for differences among observers, and one without (Fig. 2). We estimated the observer effects (and associated uncertainty about those effects) using a linear mixed model, with observer as a random effect, built in the Stan probabilistic programming language (Carpenter et al., 2017). Because observer and site are strongly related (observers tend to repeatedly sample the same site), site-level random effects were included to ensure that inferred deviations were actually observer-related (as opposed to being related to the sites that a given observer happened to see). The resulting model is described mathematically and with code in Supplemental Information 2. The model partitions the variance in observed richness values into site-level variance, observer-level variance, and residual variance (e.g., variation within a site from year to year).

Figure 2 (A) Model predictions for Pennsylvania route 35 when all observers are treated the same (black points). (B) Model predictions for the same route when accounting for systematic differences between observers (represented by the points’ colors).

In this example most models are made more robust to observer turnover. Note that the “naive” model is less sensitive to observer turnover, and does not benefit as much from modeling it.

Across our six modeling approaches (described below), we used estimates from the observer model in three different ways. First, the expected values for site-level richness were used directly as our “average” baseline model (see below). For the two models that made species-level predictions, the estimated observer effects were included alongside the environmental variables as predictors. Finally, we trained the remaining models to predict observer-corrected richness values (i.e., observed richness minus the observer effect, or the number of species that would have been recorded by a “typical” observer). Since the site-level and observer-level random effects are not known precisely, we represented the range of possible values using 500 Monte Carlo samples from the posterior distribution over these effects. Each downstream model was then trained 500 times using different possible values for the random effects.

Models: site-level models

Three of the models used in this study were fit to each site separately, with no environmental information (Table 1). These models were fit to each BBS route twice: once using the residuals from the observer model, and once using the raw richness values. When correcting for observer effects, we averaged across 500 models that were fit separately to the 500 Monte Carlo estimates of the observer effects, to account for our uncertainty in the true values of those effects. All of these models use a Gaussian error distribution (rather than a count distribution) for reasons discussed below (see “Model evaluation”).

Baseline models

We used two simple baseline models as a basis for comparison with the more complex models (Fig. 1, Table 1). The first baseline, called the “average” model, treated site-level richness observations either as uncorrelated noise around a site-level constant: yt=μ+ϵt.

Predictions from the “average” model are thus centered on μ, which could either be the mean of the raw training richness values, or an output from the observer model. This model’s confidence intervals have a constant width that depends on the standard deviation of ϵ, which can either be the standard deviation of the raw training richness values, or σresidual from the observer model; see Supplemental Information 2).

The second baseline, called the “naive” model (Hyndman & Athanasopoulos, 2014), was a simple autoregressive process with a single year of history, i.e., an ARIMA(0,1,0) model: y=yt−1+ϵt,

where the standard deviation of ϵ is a free parameter for each site. In contrast to the “average” model, whose predictions are based on the average richness across the whole time series, the “naive” model predicts that future observations will be similar to the final observed value (e.g., in our hindcasts the value observed in 2003). Moreover, because the ϵ values accumulate over time, the confidence intervals expand rapidly as the predictions extend farther into the future. Despite these differences, both models’ richness predictions are centered on a constant value, so neither model can anticipate any trends in richness or any responses to future environmental changes.

Time series models

We used Auto-ARIMA models (based on the auto.arima function in the package forecast; Hyndman, 2017) to represent an array of different time-series modeling approaches. These models can include an autoregressive component (as in the “naive” model, but with the possibility of longer-term dependencies in the underlying process), a moving average component (where the noise can have serial autocorrelation) and an integration/differencing component (so that the analysis could be performed on sequential differences of the raw data, accommodating more complex patterns including trends). The auto.arima function chooses whether to include each of these components (and how many terms to include for each one) using AICc (Hyndman, 2017). Since there is no seasonal component to the BBS time-series, we did not include a season component in these models. Otherwise we used the default settings for this function (See Supplemental Information for details).

Models: environmental models

In contrast to the single-site models, most attempts to predict species richness focus on using correlative models based on environmental variables. We tested three common variants of this approach: direct modeling of species richness; stacking individual species distribution models; and joint species distribution models (JSDMs). Following the standard approach, site-level random effects were not included in these models as predictors, meaning that this approach implicitly assumes that two sites with identical Bioclim, elevation, and NDVI values should have identical richness distributions. As above, we included observer effects and the associated uncertainty by running these models 500 times (once per MCMC sample).

“Macroecological” model: richness GBM

We used a boosted regression tree model using the gbm package (Ridgeway et al., 2017) to directly model species richness as a function of environmental variables. Boosted regression trees are a form of tree-based modeling that work by fitting thousands of small tree-structured models sequentially, with each tree optimized to reduce the error of its predecessors. They are flexible models that are considered well suited for prediction (Elith, Leathwick & Hastie, 2008). This model was optimized using a Gaussian likelihood, with a maximum interaction depth of 5, shrinkage of 0.015, and up to 10,000 trees. The number of trees used for prediction was selected using the “out of bag” estimator; this number averaged 6,700 for the non-observer data and 7,800 for the observer-corrected data.

Species Distribution Model: stacked random forests

Species distribution models (SDMs) predict individual species’ occurrence probabilities using environmental variables. Species-level models are used to predict richness by summing the predicted probability of occupancy across all species at a site. This avoids known problems with the use of thresholds for determining whether or not a species will be present at a site (Pellissier et al., 2013; Calabrese et al., 2014). Following Calabrese et al. (2014), we calculated the uncertainty in our richness estimate by treating richness as a sum over independent Bernoulli random variables: σrichness2=∑ipi1−pi, where i indexes species. By itself, this approach is known to underestimate the true community-level uncertainty because it ignores the uncertainty in the species-level probabilites (Calabrese et al., 2014). To mitigate this problem, we used an ensemble of 500 estimates for each of the species-level probabilities instead of just one, propagating the uncertainty forward. We obtained these estimates using random forests, a common approach in the species distribution modeling literature. Random forests are constructed by fitting hundreds of independent regression trees to randomly-perturbed versions of the data (Cutler et al., 2007; Caruana, Karampatziakis & Yessenalina, 2008). When correcting for observer effects, each of the 500 trees in our species-level random forests used a different Monte Carlo estimate of the observer effects as a predictor variable.

Joint Species Distribution Model: mistnet

Joint species distribution models (JSDMs) are a new approach that makes predictions about the full composition of a community instead of modeling each species independently as above (Warton et al., 2015). JSDMs remove the assumed independence among species and explicitly account for the possibility that a site will be much more (or less) suitable for birds in general (or particular groups of birds) than one would expect based on the available environmental measurements alone. As a result, JSDMs do a better job of representing uncertainty about richness than stacked SDMs (Harris, 2015; Warton et al., 2015). We used the mistnet package (Harris, 2015) because it is the only JSDM that describes species’ environmental associations with nonlinear functions.

Model evaluation

We defined model performance for all models in terms of continuous Gaussian errors, instead of using discrete count distributions. Variance in species richness within sites was lower than predicted by several common count models, such as the Poisson or binomial (i.e., richness was underdispersed for individual sites), so these count models would have had difficulty fitting the data (cf. Calabrese et al., 2014). The use of a continuous distribution is adequate here, since richness had a relatively large mean (51) and all models produce continuous richness estimates. When a model was run multiple times for the purpose of correcting for observer effects, we used the mean of those runs’ point estimates as our final point estimate and we calculated the uncertainty using the law of total variance (i.e., V ary ¯+EV ary, or the variance in point estimates plus the average residual variance).

We evaluated each model’s forecasts using the data for each year between 2004 and 2013. We used three metrics for evaluating performance: (1) root-mean-square error (RMSE) to determine how far, on average, the models’ predictions were from the observed value; (2) the 95% prediction interval coverage to determine how well the models predicted the range of possible outcomes; and (3) deviance (i.e., negative 2 times the Gaussian log-likelihood) as an integrative measure of fit that incorporates both accuracy and uncertainty. In addition to evaluating forecast performance in general, we evaluated how performance changed as the time horizon of forecasting increased by plotting performance metrics against year. Finally, we decomposed each model’s squared error into two components: the squared error associated with site-level means and the squared error associated with annual fluctuations in richness within a site. This decomposition describes the extent to which each model’s error depends on consistent differences among sites versus changes in site-level richness from year to year.

All analyses were conducted using R (R Core Team, 2017). Primary R packages used in the analysis included dplyr (Wickham et al., 2017), tidyr (Wickham, 2017), gimms (Detsch, 2016), sp (Pebesma & Bivand, 2005; Bivand, Pebesma & Gomez-Rubio, 2013), raster (Hijmans, 2016), prism (PRISM Climate Group, 2004), rdataretriever (McGlinn et al., 2017), forecast (Hyndman & Khandakar, 2008; Hyndman, 2017), git2r (Widgren and others, 2016), ggplot (Wickham, 2009), mistnet (Harris, 2015), viridis (Garnier, 2017), rstan (Stan Development Team, 2016), yaml (Stephens, 2016), purrr (Henry & Wickham, 2017), gbm (Ridgeway et al., 2017), randomForest (Liaw & Wiener, 2002). Code to fully reproduce this analysis is available on GitHub (https://github.com/weecology/bbs-forecasting) and archived on Zenodo (Harris, White & Taylor, 2017a).

Results

The site-observer mixed model found that 70% of the variance in richness in the training set could be explained by differences among sites, and 21% could be explained by differences among observers. The remaining 9% represents residual variation, where a given observer might report a different number of species in different years. In the training set, the residuals had a standard deviation of about 3.6 species. After correcting for observer differences, there was little temporal autocorrelation in these residuals (i.e., the residuals in one year explain 1.3% of the variance in the residuals of the following year), suggesting that richness was approximately stationary between 1982 and 2003.

When comparing forecasts for richness across sites all methods performed well (Fig. 3; all R2 > 0.5). However SDMs (both stacked and joint) and the macroecological model all failed to successfully forecast the highest-richness sites, resulting in a notable clustering of predicted values near ∼60 species and the poorest model performance (R2 = 0.52–0.78, versus R2 = 0.67–0.87 for the within-site methods).

Figure 3 Performance of six forecasting models for predicting species richness one year (2004) and ten years into the future (2013).

Plots show observed vs. predicted values for species richness. Models were trained with data from 1982–2003. In general, the single-site models (A) outperformed the environmental models (B). The accuracy of the predictions generally declined as the timescale of the forecast was extended from 2004 to 2013.

While all models generally performed well in absolute terms (Fig. 3), none consistently outperformed the “average” baseline (Fig. 4). The auto-ARIMA was generally the best-performing non-baseline model, but in many cases (67% of the time), the auto.arima procedure selected a model with only an intercept term (i.e., no autoregressive terms, no drift, and no moving average terms), making it similar to the “average” model. All five alternatives to the “average” model achieved lower error on some of the sites in some years, but each one had a higher mean absolute error and higher mean deviance (Fig. 4).

Figure 4 Difference between the forecast error of models and the error of the average baseline using both absolute error (A) and deviance (B).

Differences are taken for each site and testing year so that errors for the same forecast are directly compared. The error of the average baseline is by definition zero and is indicated by the horizontal gray line. None of the five models provided a consistent improvement over the average baseline. The absolute error of the models was generally similar or larger than that of the “average” model, with large outliers in both directions. The deviance of the models was also generally higher than the “average” baseline.

Most models produced confidence intervals that were too narrow, indicating overconfident predictions (Fig. 5B). The random forest-based SDM stack was the most overconfident model, with only 72% of observations falling inside its 95% confidence intervals. This narrow predictive distribution resulted in the stacked SDM having notably higher deviance (Fig. 5C) than the next-worst model, even though its point estimates were not unusually bad in terms of RMSE (Fig. 5A). As discussed elsewhere (Harris, 2015), this overconfidence results from the assumption in stacked SDMs that errors in the species-level predictions are independent. The GBM-based “macroecological” model and the mistnet JSDM had the best calibrated uncertainty estimates (Fig. 5C) and therefore their relative performance was higher in terms of deviance than in terms of RMSE. The “naive” model was the only model whose confidence intervals were too wide (Fig. 5C), which can be attributed to the rapid rate at which these intervals expand (Fig. 1).

Figure 5 Change in performance of the six forecasting models with the time horizon of the forecast (1–10 years into the future).

(A) Root mean square error (rmse; the error in the point estimates) shows the three environmental models tending to show the largest errors at all time horizons and all models getting worse as they forecast further into the future at approximately the same rate. (C) Coverage of a model’s 95% confidence intervals (how often the observed values fall inside the predicted range; the black line indicates ideal performance) shows that the “naive” model’s predictive distribution is too wide (capturing almost all of the data) and the stacked SDM’s predictive distribution is too narrow (missing almost a third of the observed richness values by 2014). (B) Deviance (lack of fit of the entire predictive distribution) shows the stacked species distribution models with much higher error than other models and shows that the “naive” model’s deviance grows relatively quickly.

Partitioning each model’s squared error shows that the majority of the residual error was attributed to errors in estimating site-level means, rather than errors in tracking year-to-year fluctuations (Fig. 6). The “average” model, which was based entirely on site-level means, had the lowest error in this regard. In contrast, the three environmental models showed larger biases at the site level, though they still explained most of the variance in this component. This makes sense, given that they could not explicitly distinguish among sites with similar climate, NDVI, and elevation. Interestingly, the environmental models had higher squared error than the baselines did for tracking year-to-year fluctuations in richness as well.

Figure 6 Partitioning of the squared error for each model into site and year components.

The site-level mean component shows consistent over or under estimates of richness at a site across years. The annual fluctuation compoonent shows errors in predicting fluctuations in a site’s richness over time. Both components of the mean squared error were lower for the single-site models than for the environmental models.

Accounting for differences among observers generally improved measures of model fit (Fig. 7). Improvements primarily resulted from a small number of forecasts where observer turnover caused a large shift in the reported richness values. The naive baseline was less sensitive to these shifts, because it largely ignored the richness values reported by observers that had retired by the end of the training period (Fig. 1). The average model, which gave equal weight to observations from the whole training period, showed a larger decline in performance when not accounting for observer effects—especially in terms of coverage. The performance of the mistnet JSDM was notable here, because its prediction intervals retained good coverage even when not correcting for observer differences, which we attribute to the JSDM’s ability to model this variation with its latent variables.

Figure 7 Controlling for differences among observers generally improved each model’s predictions, on average.

Average change in model performance from accounting for observer variation based on (A) Root mean square error (RMSE), (B) coverage, and (C) deviance. Arrows indicate the direction and magnitude of change after applying the observer model, with the base of arrow showing the value when not controlling for observer differences and the tip showing the value when controlling for observer differences. The solid line in C indicates the ideal coverage value for the 95% prediction interval.

Discussion

Forecasting is an emerging imperative in ecology; as such, the field needs to develop and follow best practices for conducting and evaluating ecological forecasts (Clark et al., 2001). We have used a number of these practices (Box 1) in a single study that builds and evaluates forecasts of biodiversity in the form of species richness. The results of this effort are both promising and humbling. When comparing predictions across sites, many different approaches produce reasonable forecasts (Fig. 3). If a site is predicted to have a high number of species in the future, relative to other sites, it generally does (Fig. 3). However, none of the methods evaluated reliably determined how site-level richness changes over time (Figs. 4 and 5), which is generally the stated purpose of these forecasts. As a result, baseline models, which did not attempt to anticipate changes in richness over time, generally provided the best forecasts for future biodiversity. While this study is restricted to breeding birds in North America, its results are consistent with a growing literature on the limits of ecological forecasting, as discussed below.

The most commonly used methods for forecasting future biodiversity, SDMs and macroecological models, both produced worse forecasts than time-series models and simple baselines. This weakness suggests that predictions about future biodiversity change should be viewed with skepticism unless the underlying models have been validated temporally, via hindcasting and comparison with simple baselines. Since site-level richness is relatively stable, spatial validation is not enough: a model can have high accuracy across spatial gradients without being able to predict changes over time. This gap between spatial and temporal accuracy is known to be important for species-level predictions (Rapacciuolo et al., 2012; Oedekoven et al., 2017); our results indicate that it is substantial for higher-level patterns like richness as well. SDMs’ poor temporal predictions are particularly sobering, as these models have been one of the main foundations for estimates of the predicted loss of biodiversity to climate change over the past two decades (Thomas et al., 2004; Thuiller et al., 2011; Urban, 2015). Our results also highlight the importance of comparing multiple modeling approaches when conducting ecological forecasts, and in particular, the value of comparing results to simple baselines to avoid over-interpreting the information present in these forecasts (Box 1). Disciplines that have more mature forecasting cultures often do this by reporting “forecast skill”, i.e., the improvement in the forecast relative to a simple baseline (Jolliffe & Stephenson, 2003). We recommend following the example of Perretti, Munch & Sugihara (2013) and adopting this approach in future ecological forecasting research.

When comparing different methods for forecasting our results demonstrate the importance of considering uncertainty (Box 1; Clark et al., 2001; Dietze et al., 2018). Previous comparisons between stacked SDMs and macroecological models reported that the methods yielded equivalent results for forecasting diversity (Algar et al., 2009; Distler, Schuetz & Langham, 2015). While our results support this equivalence for point estimates, they also show that stacked SDMs dramatically underestimate the range of possible outcomes; after ten years, more than a third of the observed richness values fell outside the stacked SDMs’ 95% prediction intervals. Consistent with Harris (2015) and Warton et al. (2015), we found that JSDMs’ wider prediction intervals enabled them to avoid this problem. Macroecological models appear to share this advantage, while being considerably easier to implement.

We have only evaluated annual forecasts up to a decade into the future, but forecasts are often made with a lead time of 50 years or more. These long-term forecasts are difficult to evaluate given the small number of century-scale datasets, but are important for understanding changes in biodiversity at some of the lead times relevant for conservation and management. Two studies have assessed models of species richness at longer lead times (Algar et al., 2009; Distler, Schuetz & Langham, 2015), but the results were not compared to baseline or time-series models (in part due to data limitations) making them difficult to compare to our results directly. Studies on shorter time scales, such as ours, provide one way to evaluate our forecasting methods without having to wait several decades to observe the effects of environmental change on biodiversity (Petchey et al., 2015; Dietze et al., 2018; Tredennick et al., 2016), but cannot fully replace longer-term evaluations (Tredennick et al., 2016). In general, drivers of species richness can differ at different temporal scales (Rosenzweig, 1995; White, 2004; White, 2007; Blonder et al., 2017), so different methods may perform better for different lead times. In particular, we might expect environmental and ecological information to become more important at longer time scales, and thus for the performance of simple baseline forecasts to degrade faster than forecasts from SDMs and other similar models. We did observe a small trend in this direction: deviance for the auto-ARIMA models and for the average baseline grew faster than for two of the environmental models (the JSDM and the macroecological model), although this difference was not statistically significant for the average baseline.

While it is possible that models that include species’ relationships to their environments or direct environmental constraints on richness will provide better fits at longer lead times, it is also possible that they will continue to produce forecasts that are worse than baselines that assume the systems are static. This would be expected to occur if richness in these systems is not changing over the relevant multi-decadal time scales, which would make simpler models with no directional change more appropriate. Recent suggestions that local scale richness in some systems is not changing directionally at multi-decadal scales supports this possibility (Brown et al., 2001; Ernest & Brown, 2001; Vellend et al., 2013; Dornelas et al., 2014). A lack of change in richness may be expected even in the presence of substantial changes in environmental conditions and species composition at a site due to replacement of species from the regional pool (Brown et al., 2001; Ernest & Brown, 2001). On average, the Breeding Bird Survey sites used in this study show little change in richness (site-level SD of 3.6 species, after controlling for differences among observers; see also La Sorte & Boecklen, 2005). The absence of rapid change in this dataset is beneficial for the absolute accuracy of forecasts across different sites: when a past year’s richness is already known, it is easy to estimate future richness. Ward et al. (2014) found similar patterns in time series of fisheries stocks, where relatively stable time series were best predicted by simple models and more complex models were only beneficial with more dynamic time series. The site-level stability of the BBS data also explains why SDMs and macroecological models perform relatively well at predicting future richness, despite failing to capture changes in richness over time.

The relatively stable nature of the BBS richness time-series also makes it difficult to improve forecasts relative to simple baselines, since those baselines are already close to representing what is actually occurring in the system. It is possible that in systems exhibiting directional changes in richness and other biodiversity measures that models based on spatial patterns may yield better forecasts. Future research in this area should determine if regions or time periods exhibiting strong directional changes in biodiveristy are better predicted by these models and also extend our forecast horizon analyses to longer timescales where possible. Our results also suggest that future efforts to understand and forecast biodiversity should incorporate species composition, since lower-level processes are expected to be more dynamic (Ernest & Brown, 2001; Dornelas et al., 2014) and contain more information about how the systems are changing (Harris, 2015). More generally, determining the forecastability of different aspects of ecological systems under different conditions is an important next step for the future of ecological forecasting.

Future biodiversity forecasting efforts also need to address the uncertainty introduced by the error in forecasting the environmental conditions that are used as predictor variables. In this, and other hindcasting studies, the environmental conditions for the “future” are known because the data has already been observed. However, in real forecasts the environmental conditions themselves have to be predicted, and environmental forecasts will also have uncertainty and bias. Ultimately, ecological forecasts that use environmental data will therefore be more uncertain than our current hindcasting efforts, and it is important to correctly incorporate this uncertainty into our models (Clark et al., 2001; Dietze, 2017). Limitations in forecasting future environmental conditions—particularly at small scales—will present continued challenges for models incorporating environmental variables, and this may result in a continued advantage for simple single-site approaches.

In addition to comparing and improving the process models used for forecasting it is important to consider the observation models. When working with any ecological dataset, there are imperfections in the sampling process that have the potential to influence results. With large scale surveys and citizen science datasets, such as the Breeding Bird Survey, these issues are potentially magnified by the large number of different observers and by major differences in the habitats and species being surveyed (Sauer, Peterjohn & Link, 1994). Accounting for differences in observers reduced the average error in our point estimates and also improved the coverage of the confidence intervals. In addition, controlling for observer effects resulted in changes in which models performed best, most notably improving most models’ point estimates relative to the naive baseline. This demonstrates that modeling observation error can be important for properly estimating and reducing uncertainty in forecasts and can also lead to changes in the best methods for forecasting (Box 1). This suggests that, prior to accounting for observer effects, the naive model performed well largely because it was capable of accommodating rapid shifts in estimated richness introduced by changes in the observer. These kinds of rapid changes were difficult for the other single-site models to accommodate. Another key aspect of an ideal observation model is imperfect detection. In this study, we did not address differences in detection probability across species and sites (Boulinier et al., 1998) since there is no clear way to address this issue using North American Breeding Bird Survey data without making strong assumptions about the data (i.e., assuming there is no biological variation in stops along a route; White & Hurlbert, 2010), but this would be a valuable addition to future forecasting models.

The science of forecasting biodiversity remains in its infancy and it is important to consider weaknesses in current forecasting methods in that context. In the beginning, weather forecasts were also worse than simple baselines, but these forecasts have continually improved throughout the history of the field (McGill, 2012; Silver, 2012; Bauer, Thorpe & Brunet, 2015). One practice that led to improvements in weather forecasts was that large numbers of forecasts were made publicly, allowing different approaches to be regularly assessed and refined (McGill, 2012; Silver, 2012). To facilitate this kind of improvement, it is important for ecologists to start regularly making and evaluating real ecological forecasts, even if they perform poorly, and to make these forecasts openly available for assessment (McGill, 2012; Dietze et al., 2018). These forecasts should include both short-term predictions, which can be assessed quickly, and mid- to long-term forecasts, which can help ecologists to assess long time-scale processes and determine how far into the future we can successfully forecast (Dietze et al., 2018; Tredennick et al., 2016). We have openly archived forecasts from all six models through the year 2050 (Harris, White & Taylor, 2017b), so that we and others can assess how well they perform. We plan to evaluate these forecasts and report the results as each new year of BBS data becomes available, and make iterative improvements to the forecasting models in response to these assessments.

Making successful ecological forecasts will be challenging. Ecological systems are complex, our fundamental theory is less refined than for simpler physical and chemical systems, and we currently lack the scale of data that often produces effective forecasts through machine learning. Despite this, we believe that progress can be made if we develop an active forecasting culture in ecology that builds and assesses forecasts in ways that will allow us to improve the effectiveness of ecological forecasts more rapidly (Box 1; McGill, 2012; Dietze et al., 2018). This includes expanding the scope of the ecological and environmental data we work with, paying attention to uncertainty in both model building and forecast evaluation, and rigorously assessing forecasts using a combination of hindcasting, archived forecasts, and comparisons to simple baselines.

Box 1: Best practices for making and evaluating ecological forecasts

1. Compare multiple modeling approaches

Typically ecological forecasts use one modeling approach or a small number of related approaches. By fitting and evaluating multiple modeling approaches we can learn more rapidly about the best approaches for making predictions for a given ecological quantity (Clark et al., 2001; Ward et al., 2014). This includes comparing process-based (e.g., Kearney & Porter, 2009) and data-driven models (e.g., Ward et al., 2014), as well as comparing the accuracy of forecasts to simple baselines to determine if the modeled forecasts are more accurate than the naive assumption that the world is static (Jolliffe & Stephenson, 2003; Perretti, Munch & Sugihara, 2013).

2. Use time-series data when possible

Forecasts describe how systems are expected to change through time. While some areas of ecological forecasting focus primarily on time-series data (Ward et al., 2014), others primarily focus on using spatial models and space-for-time substitutions (Blois et al., 2013). Using ecological and environmental time-series data allows the consideration of actual dynamics from both a process and error structure perspective (Tredennick et al., 2016).

3. Pay attention to uncertainty

Understanding uncertainty in a forecast is just as important as understanding the average or expected outcome. Failing to account for uncertainty can result in overconfidence in uncertain outcomes leading to poor decision making and erosion of confidence in ecological forecasts (Clark et al., 2001). Models should explicitly include sources of uncertainty and propagate them through the forecast where possible (Clark et al., 2001; Dietze, 2017). Evaluations of forecasts should assess the accuracy of models’ estimated uncertainties as well as their point estimates (Dietze, 2017).

4. Use predictors related to the question

Many ecological forecasts use data that is readily available and easy to work with. While ease of use is a reasonable consideration it is also important to include predictor variables that are expected to relate to the ecological quantity being forecast. Time-series of predictors, instead of long-term averages, are also preferable to match the ecologial data (see #2). Investing time in identifying and acquiring better predictor variables may have at least as many benefits as using more sophisticated modeling techniques (Kent, Bar-Massada & Carmel, 2014).

5. Address unknown or unmeasured predictors

Ecological systems are complex and many biotic and abiotic aspects of the environment are not regularly measured. As a result, some sites may deviate in consistent ways from model predictions. Unknown or unmeasured predictors can be incorporated in models using site-level random effects (potentially spatially autocorrelated) or by using latent variables that can identify unmeasured gradients (Harris, 2015).

6. Assess how forecast accuracy changes with time-lag

In general, the accuracy of forecasts decreases with the length of time into the future being forecast (Petchey et al., 2015). This decay in accuracy should be considered when evaluating forecasts. In addition to simple decreases in forecast accuracy the potential for different rates of decay to result in different relative model performance at different lead times should be considered.

7. Include an observation model

Ecological observations are influenced by both the underlying biological processes (e.g., resource limitation) and how the system is sampled. When possible, forecasts should model the factors influencing the observation of the data (Yu, Wong & Hutchinson, 2010; Hutchinson, Liu & Dietterich, 2011; Schurr et al., 2012).

8. Validate using hindcasting

Evalutating a model’s predictive performance across time is critical for understanding if it is useful for forecasting the future. Hindcasting uses a temporal out-of-sample validation approach to mimic how well a model would have performed had it been run in the past. For example, using occurance data from the early 20th century to model distributions which are validated with late 20th century occurances. Dense time series, such as yearly observations, are desirable to also evalulate the forecast horizon (see #6), but this is not a strict requirement.

9. Publicly archive forecasts

Forecast values and/or models should be archived so that they can be assessed after new data is generated (McGill, 2012; Silver, 2012; Dietze et al., 2018). Enough information should be provided in the archive to allow unambiguous assessment of each forecast’s performance (Tetlock & Gardner, 2016).

10. Make both short-term and long-term predictions

Even in cases where long-term predictions are the primary goal, short-term predictions should also be made to accommodate the time-scales of planning and management decisions and to allow the accuracy of the forecasts to be quickly evaluated (Dietze et al., 2018; Tredennick et al., 2016).

Supplemental Information

Table S1 CMIP modeling groups

Click here for additional data file.

Supplemental Information 1 Details for the observer model and auto-ARIMA model, with code

Click here for additional data file.

We thank the developers and providers of the data and software that made this research possible including: the PRISM Climate Group at Oregon State University, the staff at USGS and volunteer citizen scientists associated with the North American Breeding Bird Survey, NASA, the World Climate Research Programme’s Working Group on Coupled Modelling and its working groups, the US Department of Energy’s Program for Climate Model Diagnosis and Intercomparison, and the Global Organization for Earth System Science Portals. SKM Ernest and AC Perry provided valuable comments that improved the clarity of this manuscript.

Additional Information and Declarations

Competing Interests

Author Contributions

Data Availability

Ethan P. White is an Editor for PeerJ. We declare that there are no other competing interests.

David J. Harris, Shawn D. Taylor and Ethan P. White conceived and designed the experiments, performed the experiments, analyzed the data, contributed reagents/materials/analysis tools, wrote the paper, prepared figures and/or tables, reviewed drafts of the paper.

The following information was supplied regarding data availability:

Code to download the data & replicate the analysis:

David J. Harris, Shawn D. Taylor, & Ethan P. White. (2018, February 1). weecology/bbs-forecasting: Accepted at PeerJ (Version v1.0.0). Zenodo. https://doi.org/10.5281/zenodo.888988.

Predictions:

David J. Harris, Shawn D. Taylor, & Ethan P. White. (2018, January 26). weecology/forecasts 2018-01-26 (Version 2018-01-26). Zenodo. http://doi.org/10.5281/zenodo.839580.

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
