# Peer review of "Forecasting biodiversity in breeding birds using best practices"

_PeerJ, doi:10.7717/peerj.4278_

## Round 0.1 · original submission · Minor Revisions

· Academic Editor

Minor Revisions

The reviewers and I agree that this manuscript is interesting, timely and well-written. I will add a few of my own observations here. First, and least important - normally submitted manuscripts do not hyphenate and do not right justify.

Next, in your figure 1, which forms the basis for comparison of model accuracy over time, I really expected you to choose data WITH a temporal pattern. After all, it looks to me like you have a time series in which species richness simply varies around a more or less stable average value, and the variation can be due to nothing (random), observer bias, and any number of species population trends (and combinations thereof). Thus, you have a situation of years of no trend, and use those data to predict no trend into the future? I would have thought you would choose data with a definite time trend that was already potentially correlated with, say, climate change, and then use hindcasting and forecasting to test whether the models accurately predict a trend into the "future"! To me, it is not surprising that models do not "...successfully turn spatial data into useful temporal predictions about biodiversity at decadal time-scales." But, I wonder if you chose data WITH a trend, you would have gotten other results.

Finally, your figure two seems to reverse panels A and B, neither of which is marked in the text.

I will leave the remainder of the comments to the two reviewers.

·

Basic reporting

I found the paper very clear and easy to follow, with a good intro/discussion that tied the current work into previous studies. All data shared - in addition to code. More minor comments in General Comments section.

Experimental design

This question and forecasting competition was well designed. I had minor comments to improve clarity with models. The analysis is all documented/replicable on the Github repo. More minor comments in General Comments section.

Validity of the findings

All data is robust, and analyses are sound. Conclusions and recommendations are well stated. The authors have really done a great job with transparency - this is a model of what all papers should strive for. See additional comments in General Comments section

Additional comments

Review of Harris et al. Forecasting biodiversity in breeding birds using best practices

In general, I thought this paper was interesting and well written. These results add to the increasing literature supporting the idea that ecological forecasting in time/space is difficult. I think the recommendations/best practice bullet points described by the authors are essential for anyone interested in forecasting this kind of data. All of my points below can be considered minor issues, which I think will improve the paper’s clarity.

Introduction:
Line 41: I think this is generally more common in terrestrial ecology – in fisheries these SDM models with more of a theoretical background don’t seem to be used as much

Line 62: I really like the recommendations / clarity in Box 1, and think these will be accessible to most ecologists

Line 71-73: I’m re-reading this after commenting on data, and am now a little confused. The bird responses are annual snapshots during the breeding season? And I thought the environmental data you used was annual or quarterly, not monthly?
https://www.mbr-pwrc.usgs.gov/bbs/grass/genintro.htm

Methods:
Line 92: For those not familiar with the BBS protocol, maybe describe in a sentence. I could be mis-remembering, but isn’t each site made up of 50 stops, and presumably you aggregated richness across these stops?

Line 105: I assume all these species have been similarly ID’d over the course of the dataset – e.g. birds split into sub-species haven’t been included.

Line 107: For the environmental variables, do you have any that match up with the temporal resolution of the response data (monthly?). It seems like all the variables are annual / quarterly / etc.

Line 124: I’m not as familiar with terrestrial projections, but are the CMIP5 projections relatively unbiased / precise when compared to recent data?

Line 138: Maybe also cite any of the interfaces you used, like RStan:
http://mc-stan.org/users/citations/

Line 140: I’m curious why you didn’t consider spatial structure in the random effects here, using GMRF models or similar? One hand these models might be overkill if the habitat/environmental variables (which are spatially correlated) explain variability in richness, but the spatial correlation may pick up other sources of variability not accounted for by those variables.

Line 141-143: I think the paper thus far is pretty clear. I was hoping to see a few equations here – it helps me especially when components start to become hierarchical!

Line 163-164: again, I think equations could really help here to describe the baseline models. I think it’s also important to use these to clarify key details – for example, by training models on a site-by-site basis, you’re estimating separate parameters by site. For example with the uncorrelated noise model, you’re estimating unique variances by site right? Maybe say something like this, in addition to including equations.

Line 180 – Maybe also mention this is in the forecast package – as you do with gbm below?

Line 183: Because these defaults may change over time, it’d be good to specify things like the ranges of AR / MA order that you searched over, which criteria was used to compare models, etc. And whether or not you included a drift parameter?

Table 1 caption: without getting to more methods (below), I don’t understand the sentence ‘Environmental models were trained using all sites together, without information regarding which transects occurred at which site or during which year’.

Table 1: Just my personal preference, but I’d include the check marks and replace the ‘Xs’ with blank cells

Line 194-221: It seems for each approach you only considered the full models with all environmental covariates? I’m curious whether any of the covariates appeared to not improve explanatory power, and whether you considered models that dropped these uninformative predictors?

Line 216: Cite randomForest / caret/ party R package – or whatever else you used?

Line 242: Again, I think including the equation here would be useful

Line 245: In our 2014 forecasting paper, we used MASE over RMSE because of rationale in Hyndmand and Koehler (2006). Using RMSE here is probably ok because you probably don’t have zeros/huge values/etc

Line 249: Can you elaborate on what you did with the deviance? By itself mean deviance is just a measure of average fit (I think most of the models are giving you this). Var(deviance) should incorporate precision – but I’ve only seen this extracted from Bayesian models

Line 266: I think you guys have done a great job putting together this public repository + made the analysis totally transparent. My one hang up with the repo was that it’d be awesome to include a short vignette with example data from this project that is more easily accessible to readers without having to install Jupyter/R packages/etc. For example – I was travelling when I started this review and couldn’t install one of the R packages on my government computer without making an appointment with our IT staff (so it took ~ a day). This isn’t your fault of course, but much harder for me to dig through the repo.

Line 269: See my above comments about spatially correlated random effects / GMRF models. One way to confirm these other approaches are overkill would be to make some maps of the estimated random effects, and fit some simple models to estimate variogram parameters – e.g. even gls() in R can do this

Line 276: Is one of the potential reasons that richness is stationary that presence/absence on the BBS varies less than other measures – maybe abundance, or other measures of diversity are more variable?

Line 294: Is the narrow predictive distribution for the stacked SDM also why this exhibits the strongest trend in decreasing coverge as a function of forecast length (Fig 5)?

Line 329 – maybe reword, you use forecast 3x in this sentence

Line 414: Agreed – doing this same kind of forecasting competition with compositional data, or other species diversity measures (Simpson’s) would be interesting

Line 461: Another thread with conservation implications is that back in the 1990s, it was pretty common for single species PVA models to be forecasted 100-200 years. This seems less common, in part because some predictions may have not performed well, and others are on a time horizon that can’t be evaluated. In Steve Beissinger’s PVA book he cites Goldwasser (2000) “Variability and measurement error in extinction risk analysis...” for this point (p 133).

Reviewer 2 ·

Basic reporting

see general comments

Experimental design

see general comments

Validity of the findings

see general comments

Additional comments

In this paper, the authors use the incredibly rich data from the Breeding Bird Survey to examine how well various models perform at predicting bird richness at sites throughout North America. They train models on data from 1982-2003 and attempt to predict data from 2004-2013. They examine the performance of six different model types, two of which are null or baseline models.

I find it a bit surprising that the authors think that any of these richness models could predict a high level of accuracy for theses ten years. I think of richness estimations as a very coarse technique for looking at broad shifts in richness over relatively long time periods rather than these 5-species (~5-7%) shifts in richness on annual bases indicated in Figure 2. The authors don’t provide a lot of their raw data on sites other than as examples, but my impression is that they are pursuing an unrealistic level of accuracy and detail. That being said, given what the paper does, it’s okay. The models seem to be performed well. The authors give all of the keywords to indicate that they know what they are doing.

Here are some comments:
- Birds are not the best system for doing any sort of distribution analyses. They are incredibly vagile, hard to detect, and the distributions of breeding birds are affected by many components besides environmental variables (particularly interspecific interactions). No assessment of detectability is included here other than accounting for observer identity. However, I do realize that BBS data are detailed and compelling data to use given what a large amount of detailed time-series it provides.
- Abstract: The authors forecast the data and then test it using known data. Hindcast only refers to back-in-time predictions.
- 25-29: rather than relay the type of information that the paper will provide, the authors should summarize their results and findings
- 57-58: this is not the definition of hindcasting. This is the definition of testing. Also, the authors exclude the many deeper-time hindcasting studies that have been performed on these types of models.
- Methods: the authors perform their analyses on data that are collected from an incredibly dynamic time period. I wouldn’t expect species to have settled into a richness pattern given the incredibly disrupted nature of the last 32 years. Some strategies to try to handle these issues might be to train and predict at different temporal scales and to both forecast and hindcast the predictions. (e.g. train on 2004-2013 and try to predict 1982-1992 data + 1993-2003 data). If the authors were to make splits based on expected periods of dynamism, they could better tell whether the drivers that are included in the models are accurate predictors.
- 124-130: I didn’t see where future projections were performed/used in this paper. Why is this here?
- Fig. 2: Panels not labeled. No key for colors. Somewhat confusing.
- Fig. 6 caption: “compoonents”
- 466: only other mention of future forecasts… not sure why this is included in this paper.

---

## Round 0.2 · accepted · Accept

· Academic Editor

Accept

This version of your manuscript seems to have addressed all the issues by the reviewers and myself, however, I found a typographical error and had a few suggestions and miscellaneous observations that may help improve clarity in the text - see the attached PDF. These issues can be edited while in Production.